# Scalable OneM2M IoT Open-Source Platform Evaluated in an SDN Optical Network Controller Scenario [note 1]

**DOI:** 10.3390/s22020431

**Published:** 2022-01-07

**Authors:** Martina Troscia, Andrea Sgambelluri, Francesco Paolucci, Piero Castoldi, Paolo Pagano, Filippo Cugini

**Affiliations:** 1Institute of Communication, Information and Perception Technologies, Scuola Universitaria Superiore Sant’Anna, Via G. Moruzzi, 1, 56124 Pisa, Italy; andrea.sgambelluri@santannapisa.it (A.S.); piero.castoldi@santannapisa.it (P.C.); 2National Laboratory of Photonic Networks and Technologies, Consorzio Nazionale Interuniversitario per le Telecomunicazioni, Via G. Moruzzi, 1, 56124 Pisa, Italy; francesco.paolucci@cnit.it (F.P.); paolo.pagano@cnit.it (P.P.); filippo.cugini@cnit.it (F.C.)

**Keywords:** IoT, SDN, OneM2M, OCEAN, Mobius, disaggregation, optical network, ONOS, OpenDaylight, OpenROADM, controller, open standard

## Abstract

Software Defined Networking represents a mature technology for the control of optical networks, though all open controller implementations present in the literature still lack the adequate level of maturity and completeness to be considered for (pre)-production network deployments. This work aims at experimenting on, assessing and discussing the use of the OneM2M open-source platform in the context of optical networks. Network elements and devices are implemented as IoT devices, and the control application is built on top of an OneM2M-compliant server. The work concretely addresses the scalability and flexibility performances of the proposed solution, accounting for the expected growth of optical networks. The two experiment scenarios show promising results and confirm that the OneM2M platform can be adopted in such a context, paving the way to other researches and studies.

## 1. Introduction

Software Defined Networking (SDN) represents a mature technology for the control of optical networks. The deployment of proprietary SDN implementations in several production networks is nowadays a straightforward activity. What is essentially missing are truly effective open-source solutions that are extensively tested and documented. However, the need for disaggregation has significantly pushed the introduction of open SDN solutions, at both the node [1,2,3] and the controller levels [4,5,6,7]. In particular, two main initiatives are working on open-source implementations of SDN Controller solutions with a specific focus on optical networks. The first initiative, under the umbrella of the Linux Foundation, has developed the OpenDaylight SDN Controller [4]. Originated in 2013, it has now reached its 13th release. OpenDaylight was designed to be a broad, general-purpose platform to support a wide range of use cases, such as dynamically optimizing the network based on load and state or managing the network and multiple controllers from a central entity. Several proprietary SDN controller implementations make use of the results produced by OpenDaylight. The second initiative, as part of the Open Networking Foundation (ONF), has developed the Open Network Operating System (ONOS) SDN Controller [5]. Originated in 2014, it has been widely adopted in collaborative projects and research studies. Although being valuable solutions for testing and research purposes, both open controller implementations still lack the adequate level of maturity and completeness to be considered for (pre)-production optical network deployments. One of the reasons behind this might be related to the limited (optical) community that is dedicating effort to develop and test software contributions targeting production deployments.

In parallel with such initiatives, other open communities have developed relevant open-source solutions targeting different networking contexts. Among these initiatives, OneM2M [8,9] is a global partnership project founded in 2012 and constituted by several world’s leading ICT standards development organizations, including ETSI (Europe), ATIS and TIA (USA). The goal of the organization is to create a global technical standard for interoperability concerning the architecture, API specifications, security, and enrolment solutions for Machine-to-Machine (M2M) and Internet of Things (IoT) technologies [10,11]. OneM2M has driven a large and active open-source initiative named OCEAN, providing open-source implementations for OneM2M components, including servers (called Mobius), gateways, device platforms, self-conformance testing tools, and applications. Multiple complex IoT systems, such as vehicular networks [12], adopted the OneM2M standard to produce open and interoperable solutions.

Taking into account all the previous considerations, the idea of our work was born to combine the results of the studies on SDN Controller solutions with the usage of a platform compliant with the OneM2M standard. In this paper, we aimed at experimenting, assessing, and discussing the use of the OneM2M OCEAN open-source platform in the context of optical networks. Network elements and devices, such as Reconfigurable Add-Drop Multiplexers (ROADMs), transponders, amplifiers, and monitors, are implemented as being IoT devices. In addition, we built the optical network controller as an IoT server platform that collects information from the network elements and uses the knowledge of the network status to reconfigure the devices. Particularly interesting for us was to consider the scalability performances of the proposed solution, accounting for the expected evolution of optical networks, where the number of devices and monitors might remarkably increase along with the rate of their generated updates. More in detail, we can summarize the contributions brought by our work as follows: (i) the adoption of an OneM2M platform to control the optical networks, treating them analogously to sensor networks; (ii) the porting of NETCONF-based functionalities to a REST-based approach according to the OpenROADM paradigm, following the OneM2M standard; (iii) the adaptation of fundamental SDN functionalities for the configuration and the monitoring of optical devices in an OneM2M platform; (iv) the experimental assessment of the proposed solution with real commercial optical devices and Docker-based emulated agents.

The remainder of this paper is organized as follows. Section 2 provides an overview of the actual landscape for the control of Software Defined Networks and the applications employing the OneM2M platform. Section 3 extensively describes the OneM2M standard and the characteristics of the OneM2M Mobius server that make it suited for the chosen cases of study, together with the presentation of the experimental scenarios. Section 4 shows the results of our experiments and provides some insight into the work. A discussion of the obtained results is presented in Section 5. Finally, Section 6 contains some final remarks.

## 2. Related Work

The monitoring and control of optical networks is not a new topic in the literature. Software Defined Networking represents a mature technology to control optical networks and is broadly used in many proposed solutions. Elbers and Autenrieth [13] in 2012 highlighted the advantages of software-defined optical networks. They argued about the technologies enabling this new paradigm of control, reviewing the emerging applications and presenting new challenges of SDN Optical Networks.

The focus of the works in the literature is always on how to implement the controller. Casellas et al. [14] proposed a multi-controller solution to face some limitations in terms of either scalability, complexity or interoperability found in some other implementations. They suggested an SDN orchestration adopting multiple controllers, commonly arranged in hierarchical or flat configurations. Ramaswami et al. [15] focused on a solution for a distributed network control. They proposed a distributed network protocol for setting up, taking down, and maintaining the state of the connections in routed optical networks. Liu et al. [16] proposed an experimental validation of hierarchically-controlled Software Defined Networks architectures. Suzuki et al. [17] demonstrated how their developed path computation-capable Network Management System (NMS) could control an all-optical mesh network. The developed NMS could manage topology and physical impairments, and create GMPLS-based lightpaths according to path computation results. Paolucci et al. [18] showed that a stateful controller including a specific extension for path computation enabled a centralized controller to coordinate complex multi-step provisioning and elastic operations, such as spectrum slot expansion and hitless defragmentation. Channegowda et al. [19] introduced a unified control plane architecture based on OpenFlow for optical SDN tailored to cloud services. They adapted the architecture requirements to emerging optical transport technologies.

Recently, a remarkable number of works have addressed the capabilities and the potentials of SDN controllers to enable disaggregation at the optical layer. Kundrat et al. [20] showed an automatic provisioning workflow over a multi-vendor optical line system and a Cassini transceiver that included the optical impairment estimation phase, using the ONOS Controller. Giorgetti et al. [5] conducted an overall analysis of the ONOS performances related to the optical connection provisioning and restoration in a 7-node disaggregated network scenario, highlighting the issues of the ONOS Controller regarding the recovery reaction. Sgambelluri et al. [2] proposed a telemetry-assisted mechanism for the optimal selection of the operational mode at the controller in a multi-vendor disaggregated optical network. All the aforementioned works provide an evaluation analysis in real testbeds of limited size, highlighting that the controller is not designed for large topologies and may suffer from scalability issues, especially in the case of link failures affecting multiple optical connections simultaneously.

The literature is full of examples of optical network control through SDN while, on the other hand, it is not simple to find examples of usage of an OneM2M-compliant controller for the same purpose. However, OneM2M platforms and servers have been successfully employed in several IoT domains such as health, agriculture, vehicular networks, smart homes, smart offices. Fattah et al. [21] showed how they could build services for ageing people on top of a standard-based OneM2M platform, exploiting the information collected by heterogeneous IoT products. Rubì et al. [22] presented an Internet of Medical Things (IoMT) platform for pervasive healthcare that ensured interoperability, reliability, and scalability in an M2M-based architecture. This platform showed promising results when processing high volumes of data and allowed to extract knowledge from the collected information and build the required services. Kim et al. [23] studied a Disease Prediction System for smart farming that employed an OneM2M platform for handling the collection, analysis and prediction of agricultural environment information. The proposed Farm-as-a-Service integrated system supported high-level application services by operating and monitoring farms and managing associated devices, models and data. Jeong et al. [24] developed a smart city platform for the collection and analysis of data to generate insights for solving smart city interoperability issues, especially at a data level. Yun et al. [25] presented an implementation work of sensing and actuation capabilities for IoT devices using the OneM2M standard-based platforms. They mainly focused their study on the importance of a well-designed middleware solution compliant with the standard, providing a best practice for other platforms. Araujo et al. [26] proposed OneM2M architectures that may be used in the context of ETSI C-ITS compliant safety applications and presented some test results obtained from the implementation of these architectures with real hardware. They focused their attention on the latency introduced by the OneM2M platforms, which must be very small in the context of car-following applications, confirming that they could reduce the delay with well-designed architectures. Finally, Choi et al. [27] presented an OneM2M standard platform architecture design and its implementation supporting real-time data delivery for a drone management system. In their work, they analysed the end-to-end delay performances in various network environments.

M2M platforms following the OneM2M standard offer many interesting characteristics for applications such as interoperability, multi-domain communication, modularity and openness. These characteristics seem to also be suited for the control of optical networks. Swetina et al. [28] presented a snapshot of the latest progress in OneM2M standardization, focusing on the agreed architecture, the candidate protocols, the security aspects, and the device management and abstraction technologies.

## 3. The OneM2M IoT Platform and Proposed Implementation for Optical Network Control

### 3.1. The OneM2M Standard

Among the standard development organizations that have been working on IoT, the OneM2M Global Initiative [29] is an international partnership project whose objective was to cooperate in the production of a globally applicable, access-independent M2M layer. This initiative addresses new requirements to support interoperable and scalable IoT solutions. OneM2M promotes a global standard in contrast to national variants or proprietary approaches [30].

The OneM2M standard defines an architectural framework [31] based on a middleware technology that lies in the horizontal layer between IoT applications and a lower layer of communication networks and connected devices. The middleware layer provides a rich set of functions to manage end-to-end IoT systems. The key role of the IoT middleware is that it guarantees interoperability, and it is achieved in the OneM2M standard at a service level, providing common functions (e.g., data management, subscription/notification) to applications in different domains. The middleware also provides an abstraction layer that hides the complexity of underlying hardware devices and exposes standard Application Programming Interfaces (APIs) used by IoT applications to access the data of interest. Unlike traditional database management systems, where data are organized into schemas and queried through specific languages, each IoT device has its own data retrieval method. So the interoperability middleware eases the usage of the data by the application developers.

The OneM2M adopts a resource-oriented model for the supported services and data. A Uniform Resource Identifier (URI) uniquely addresses a resource, and the interaction between the platform and the resources happens through the basic four CRUD (Create, Retrieve, Update, Delete) operations. The OneM2M-compliant systems manage their resources hierarchically, as shown in Figure 1. The root of the hierarchy is the Common Service Entity (*CSE*) Base that represents an instantiation of a set of common service functions, such as data storage and sharing with access control and authorization, event detection and notification, and location services. Each CSE Base contains one or more Application Entities (*AE*), implementing the application service logic. AEs are organized into Containers (*CON*). The address of a resource is represented as a hierarchical address that reflects the resource structure. Considering for example Figure 1, the address of *CON1* will be *<CSE Base>/AE1/CON1*.

The OneM2M standard also specifies protocol bindings with the underlying delivery protocol, including the Hypertext Transfer Protocol (HTTP) [32], Constrained Application Protocol (CoAP) or Message Queue Telemetry Transport (MQTT), as well as a service level protocol and a service message layer format. The standard provides two kinds of methods for querying the resources: request/response or subscription/notification. In the first method, a piece of software sends an HTTP GET request to the IoT server with the Uniform Resource Locator (URL) linked to the resource it wants to get. The IoT server sends back the HTTP response containing the JSON (JavaScript Object Notation) or XML (eXtensible Markup Language) body, which corresponds to the value of the required resource. The second method involves the subscription by interested parties to a container. The IoT server will notify its subscribers of any event (e.g., a new content instance) under the subscribed resource. The subscribe/notification messages travel into HTTP POST requests.

### 3.2. The Motivation for an OneM2M-Based SDN Optical Controller

Optical networks currently employ a number of SDN controllers not specifically designed for high scalability performance. However, the transition from legacy optical networks to open and disaggregated paradigms poses the challenge to employ control elements with improved scalability for several reasons. Firstly, the number of different nodes is increasing (e.g., a single optical node is now split into the open ROADM device, the OLS device and the xPonders devices, each one exposing a different SDN agent model). Secondly, disaggregation requires more flexible and automatic commissioning and decommissioning procedures for disaggregated components. Finally, differently from legacy and old GMPLS-enabled agents, disaggregation requires handling different device abstractions (i.e., YANG models) and the related southbound APIs. IoT controllers can properly address these novel requirements in terms of the number and type of devices. Moreover, their internal design is compatible with the implementation of match-action flow rules, as required by the SDN paradigm.

The SDN Controller is the control-plane entity in charge of controlling all the network devices in the data plane. From this perspective, two are the key features of an SDN controller: the capability to perform the configuration of the traversed devices (according to the forwarding/routing decisions) and the capability to monitor, in real-time, the main parameters of all the devices (to maintain both the up-to-date network topology view and the state-full dataset of the configured connections). When considering optical networks, each device includes many parameters to be maintained (i.e., the input and output power levels of all the optical lines, the coherent parameters of each transponder port, the configuration/state of each amplifier). When the number of devices and the number of parameters grow, it is convenient to adopt distributed platforms, where configuration and monitoring functions are decoupled. From one side, the distributed approach improves the system efficiency and performance, reducing the possibility of overloading the SDN controller. From the other side, it can lead to continuous interaction among the configuration and monitoring platforms with the loss of fundamental pieces of information. The idea to adopt an IoT platform to control the optical network has been considered by exploring alternative solutions. The key functionalities of the SDN Controller perfectly fit with the basic capabilities of such a platform. By considering the optical devices as sensors, there is the possibility to send specific messages to the nodes (i.e., send configuration messages that include the full view of a node). The reception of periodic updates allows maintaining an outlook of the entire network topology and the state of the connections. For the optical network nodes, specific representation/abstraction models must be considered (i.e., OpenConfig or OpenROADM).

### 3.3. The OneM2M Platform for an Optical Network

The general concepts presented in the OneM2M standard (and summarized in Section 3.1) have been applied in our specific case. Figure 2 shows the reference scenario that we have built to run the experiments of our work. It includes an optical network encompassing transponders, ROADMs, EDFAs and other optical devices and sensors such as power monitors and low-cost OSAs. The network elements are provided with the proposed software modules supporting (i) RESTful APIs compliant with the OneM2M standard and (ii) JSON descriptions compliant with the OpenROADM standard [33]. Then, the scenario includes an optical network controller based on the OneM2M Mobius platform [34].

The OneM2M Mobius platform is an open-source IoT server platform based on the OneM2M standard. Mobius provides the needed service functions, such as registration, data management, subscription/notification, and security, according to the OneM2M specifications. In this implementation, the Mobius server communicates with the devices through RESTful APIs using GET/POST messages with a JSON body. The messages contain the set of predefined fields that are required by the OneM2M standard for the correct communication with the Mobius server. According to the standard, as explained before, the resources are organized into a hierarchical structure. Any data from a specific device is always put in the same container, increasing the number of content instances. The Mobius platform also supports the publish/subscribe paradigm and can send updates to devices and applications that have made a subscription on a container. Such Mobius platform features are relevant in the context of optical networks. It is possible to build specific applications on top of the platform to control the network. In particular, it is possible to read the status of the optical devices retrieving the information from the containers and publish specific updates in the network accordingly.

Optical devices, like most IoT devices, cannot send structured messages in the format required by the OneM2M standard. Device agents must guarantee the communication between these devices and the platform, acting as proxies. Agents can communicate with the optical devices using their specific protocols and send messages to the OneM2M platform in the required format. The possibility to manage so easily low-cost devices is very captivating for next-generation optical networks. In fact, in this way, it is possible to deploy a significantly large number of monitors and sensors and improve the awareness of Artificial Intelligence (AI)-based control and management solutions (e.g., low-cost optical spectrum analyzers—OSAs, sensors for in-fiber security intrusion detection, power monitors and optical time domain reflectometers—OTDRs—for accurate failure localization).

The network scenario relies on both real and emulated components. Real devices enhanced with the proposed software modules include two commercial 100 Gb/s polarization-multiplexed quadrature phase-shift keying (PM-QPSK) coherent transponders, one Lumentum white box ROADM, and selected additional monitors such as power monitors and low-cost OSAs. We also include a number *N* of *emulated* ROADMs. They encompass the proposed software modules while no underlying hardware is present. In the experiments, we took realistic data from the real components. The Mobius OneM2M platform acting as a basis for the optical controller has been implemented on a virtual machine running in a Linux server (Intel(R) Xeon(R) CPU E5-2650 v3 @ 2.30 GHz, 4 CPUs 4 cores, 4GB RAM). The emulated environment consists of a Docker domain, where each container emulates a network device, according to the OpenROADM data model. The communication between the emulated devices and the OneM2M platform happens through a custom network with an IPv4 network addressing scheme. However, the considered data model fully supports also the IPv6 network addressing scheme.

Each Docker container emulates a network device by leveraging two Python scripts. The first Python script implements the monitoring functionalities. In particular, the script pushes towards the OneM2M platform periodic updates related to all the main components of the node (each update message includes around 90 ports, with values mapping the real equipment in the optical laboratory). The update interval can be configured, regulating the rate of the update messages’ generation. The second Python script has been adopted to test the configuration functionality of the system, meaning the ability to receive and log the configuration messages received from the OneM2M platform. Additional details regarding the considered scenarios will be provided in the following sections.

The work focuses on two main experiments. The goal of the first experiment is twofold: (i) assess the capability of the network devices to efficiently provide parameter status updates to the controller in real-time; and (ii) assess the scalability performance of the Mobius OneM2M platform to efficiently handle a large number of network devices and an increasing number of update messages. In the second experiment, the goal is to assess the capability of the Mobius OneM2M platform to reconfigure network elements, which means: (i) to assess the capability of the platform to configure network elements for provisioning purposes; and (ii) to assess the reactivity of the network when it has to change its working parameters after receiving updates (e.g., a connection link that cannot be reached anymore).

As explained in Section 3.2, the performance metrics *R_POST_* and *T_C_* have been considered to map the two basic functionalities of an SDN controller (i.e., configuration and monitoring). More specifically, considering optical-network control, it is common to evaluate the performances of the proposed controller procedure in terms of control-plane execution time (i.e., *T_C_*), as shown in [2,5]. It is also common to evaluate the load due to the update messages received from the network devices (i.e., *R_POST_*), mapping the monitoring aspects of the SDN methodology.

In general, optical networks are employed in urban scenarios. An optical metro network is designed to aggregate metro-access rings and provide local traffic to selected metro-core nodes. For this reason, for the experimental assessment, the typical dimension of such networks has been considered, identified in a full/partial mesh of up to 30 nodes [35] or at the varying of the regional population density [36].

#### 3.3.1. Implementation of the First Use Case

In the first experiment, the controller receives periodic updates on the monitored parameters from the network elements (both real and emulated), including the measured input/output power on an ROADM port and pre-Forward Error Connection (pre-FEC) Bit Error Rate (BER) on a transponder. In particular, leveraging on specifically designed Python scripts, *N* ROADM agents send updates to the network controller (implemented through the OneM2M platform) at a configurable pace. In this experiment, we measured the rate $R_{POST}$ of POST messages per second that are successfully received and elaborated by the Mobius platform.

The usage of an OneM2M server requires building a model of the real devices it has to handle. The model for the ROADM devices has been automatically derived from real devices and saved into a JSON file. Figure 3 shows an example schema representing the internal structure of an ROADM device and its connections. In Figure 4, it is possible to see a piece of the device configuration file representing that schema.

The OneM2M server has been organized such that every container contains all the updates regarding one emulated ROADM device. Every update represents a new content instance in that container. In this way, the controller has a complete overview of the status of the network.

Python scripts emulate the behaviour of the ROADM agents. Listing 1 shows their pseudo-code. The ROADM agents periodically send updates regarding the status of the device they control. In particular, the agents detect the actual transmitting power or receiving power (depending on the port direction) of the devices’ ports into the circuit packs and communicate it to the controller. The controller can take decisions accordingly.

We repeated the experiment 30 times per parameters configuration, varying the sleep time between consecutive update submissions, the maximum number of messages each ROADM agent sends, and the number of sending agents. The objective was to observe the behaviour with different network loads and study how efficient was the solution in every condition, increasing the confidence of the produced results.

**Listing 1.** Pseudo-code of one of the *N* ROADM agents.**while** (True): **for every** circuit pack:  **for every** port:   **if** port-direction ==“tx”:    <read tx power from device port>   **else if** port-direction ==“rx”:    <read rx power from device port> <prepare message for the OneM2M> <send message> <sleep(*sleeping-time*)>

#### 3.3.2. Implementation of the Second Use Case

In the second experiment, the implemented application identifies *M* out of the *N* nodes to be configured to receive updates regarding the status of the connections with other devices (e.g., ROADM cross-connections). These nodes make a subscription with the Mobius OneM2M platform and receive messages every time the network controller detects a change in the link status. In this experiment, we measured the time $T_{C}$ experienced between the moment when the configuration is enforced at the Mobius Controller and the moment when all configuration messages are received by the subscribed ROADM agents interested in the update.

In this second experiment, it was also necessary to build a model of the optical cross-connections among nodes for the OneM2M server. Figure 5 shows this model. The structure follows the OpenROADM model hierarchy. More specifically, the configured optical cross-connection is enabled from the input Network Media Channel, configured on degree 1, to the output Network Media Channel, configured on degree 3, using the frequency 193.6 THz and setting as launch power 0 dBm. The bandwidth allocation to the optical signal is configured both at Media Channel and Network Media Channel levels (respectively sent for configuring logical interfaces for both input and output ports).

The working scenario is more elaborate in this second use case: we considered a network of ROADM devices where each device knows the other neighbour devices. In this perspective, each ROADM agent can subscribe to the OneM2M server to receive updates regarding the status of the connections that concern it. Listing 2 shows the modified pseudo-code for the ROADM agents.

**Listing 2.** Pseudo-code of one of the *M* ROADM agents.**main**():... **for every** connection with other ROADM devices:  <subscribe to updates for connection status on the OneM2M > **while** (True):  **for every** circuit pack:**for every** port:   **if** port-direction ==“tx”:    <read tx power from device port>   **else if** port-direction ==“rx”:    <read rx power from device port>  <prepare message for the OneM2M>  <send message>  <sleep(*sleeping-time*)>  **newSubscriptionUpdate**(): <get the body of the update> <process the received information> <compute $T_{C}$>

The controller can exploit the information collected from the ROADM agents sending periodic updates. The controller has become more complex: depending on the status of the ports of an ROADM device and which is the port that transmits or receives power, it knows which connections can still be used. It publishes updates on the OneM2M server about the malfunctioning links, and all the interested ROADM agents receive notifications. In this way, the ROADM devices can decide how to tune the functioning parameters according to the updates they receive through the agents. Listing 3 shows the pseudo-code for the controller. The Unified Modeling Language (UML) sequence diagram in Figure 6 shows the interactions among the different components of the system, assuming that ROADM Agent 1 is connected to ROADM Agent 2 through Connection *A* and the controller detects a malfunctioning of ROADM Agent 2.

**Listing 3.** Pseudo-code of the controller.**while** (True): <query the OneM2M for the status of ROADM devices> **for** every device:  **if** malfunctioning is detected:   <see connections of device>   **for every** broken connection:    <prepare message for the OneM2M>    <send message>

As the objective of the second use case was to study the time it takes for an update to travel from the controller to the ROADM agents, the Python scripts that emulate the agents’ behaviour have been put on different virtual machines. In this way, it was possible to take into account the network latency due to message routing.

Each experiment was repeated 30 times per parameters configuration, varying the number *M* of devices that subscribe to connection updates and the frequency and number of messages with status updates, as in the previous use case. Then the mean configuration time in each repetition has been computed.

## 4. Results

As previously explained in Section 3.3, for the first experiment, we have chosen the rate $R_{POST}$ as the performance metric, that is the number of POST messages per second that are successfully received and elaborated by the Mobius OneM2M platform.

The graph of Figure 7 shows the trend of the rate for a varying number of sending nodes (*N*), depending on the sleeping time that has been chosen between consecutive bursts of update messages. With N=100 ROADMs sending POST messages with optical parameter updates, an average value of $R_{POST}>18$ messages has been achieved (more than 65,000 messages in one hour of testing). With N=1000 ROADMs, we measured a rate $R_{POST}>5$ messages per second. When the sleeping time increases, the number of messages sent to the OneM2M decreases, and the same obviously happens to the rate $R_{POST}$. The effect is more evident for smaller *N*, as the sleeping times occur with a greater frequency. However, the trend in the graphs for the different sleeping times remains almost the same. Table 1 shows the 25th percentile, the median and the 75th percentile for the produced results.

The results confirm the capability of the network devices to efficiently provide parameter status updates to the controller in real-time, as the updates arrive with no significant delays. The network behaves as expected even when the number of sending nodes (and consequently the number of messages in the network) increases. Moreover, the moment of the day (work hours or nights) or the day of the week (working days or weekends) do not influence the results. The experiment also confirms the efficiency of the OneM2M in handling numerous network devices and an increasing number of update messages, obtaining satisfactory scalability performance. The rate of messages per second remains of the same order of magnitude.

In the second experiment, we evaluated the configuration time $T_{C}$ elapsed between the instant when the status of a connection is enforced at the Mobius Controller and the instant when all configuration messages are received by the subscribed ROADM agents interested in the update. Figure 8 shows the configuration time, varying the number *M* of nodes subscribed to configuration messages (e.g., the nodes that receive the configuration messages from the OneM2M platform) and the sleeping time that has been chosen between consecutive update messages from the *N* nodes of the network (e.g., all the nodes present in the network, that in background send update messages to the OneM2M platform).

For example, with N=25, M=5, and ROADMs generating the status message every 30 s, an average value $T_{C}$ in the order of 75 ms is achieved. With N=25, M=10, $T_{C}$ is in the order of 120 ms. The configuration time $T_{C}$ increases marginally when the sleeping time between consecutive update messages decreases, to a high update transmission rate. For example, with N=25, M=5, and ROADMs generating a message every 2 s, we obtained an average value $T_{C}$ in the order of 87 ms. Considering that we achieved a $T_{C}$ of 75 ms with a sleeping time of 30 s, the difference between the two values is minimal. The graph also confirms that the time required to enforce the configuration at the ROADM agents remains the same order of magnitude when the number of update messages in the network increases (e.g., bursts of update messages are sent more frequently as the sleeping time is decreased). Table 2 shows the 25th percentile, the median and the 75th percentile for the produced results.

The results confirm the capability of the platform to configure the network elements for provisioning purposes, as all update messages correctly arrive at the OneM2M platform from the controller and are delivered to the interested ROADM agents. The experiment also shows that the network is very reactive in changing its working parameters after receiving updates from the controller (e.g., a connection link that cannot be reached anymore). The required time is small even when the number of nodes sending (*N*) and receiving (*M*) updates increases.

Moreover, we compared the obtained results with those obtained in analogous works employing pure SDN control. It is possible to notice that the time required by the SDN controller to configure the data-plane with OpenFlow and NETCONF protocols takes around 100 ms to complete the configuration procedure (i.e., the configuration of 8 devices), considering the control-plane setup time reported in [5]. The results obtained in our study perfectly match with literature SDN-based solutions also when the number of controller devices grows.

We evaluated the scalability of the solution also considering the CPU and memory usage in the Virtual Machine hosting the OneM2M platform and on the Virtual Machines where the ROADM agents run. As shown in Figure 9, the platform can handle a significant amount of messages: the mean CPU usage is around 15%, with maximum peaks of around 50% at the moment in which the burst of messages arrives. The RAM results to be mostly unused as it has 41.4% of space. Figure 10 shows that the Virtual Machines in which the agents run have a workload similar to the one dedicated for the OneM2M platform.

## 5. Discussion

In this work, we have shown the first implementation of an IoT open-source platform applied to the management of optical networks. Network elements are implemented as IoT devices, and the optical network controller is built on a scalable IoT server platform conform with the OneM2M specifications. The high scalability of the OneM2M platform allows coping with the expected evolution of optical networks where the number of optical devices and monitors might increase together with the rate of their provided updates. The results of the two experiments show satisfactory scalability performances for the OneM2M in acquiring data from the optical network. The first scenario proves that the devices can easily share status updates with the OneM2M platform. There are no message losses, and the platform manages to handle all the received information, even when the number of nodes increases. The second scenario shows that the time needed to perform multiple simultaneous configurations increases only marginally with the number of configured elements. The timely arrival of the updates allows the controller to build a real-time snapshot of the status of the network. In this way, the network devices result to be responsive in reconfiguring themselves after learning that they cannot use some communication links anymore.

In addition, this work also shows the potential of radically different solutions designed for IoT here applied to optical networking. This aspect is rather new in literature, where most of the proposed solutions employ Software Defined Networking without an efficient software controller. The OneM2M platform offers adequate flexibility for building custom controllers that can be configured for carrying out even very complex tasks of control and monitoring. It is just a matter of developing the appropriate applications that have to run on top of the OneM2M server. The OpenROADM paradigm makes it easy to build models for the optical devices so that they can be treated like any other sensor device by the OneM2M platform. At the same time, the porting of the NET-CONF functionalities to the REST approach is simple by combining the OpenROADM paradigm and the OneM2M standard.

## 6. Conclusions

The expected evolution of optical networks, where the number of devices is constantly growing, calls for efficient solutions for the control and monitoring of these networks. Software Defined Networking is a mature technology for this scope, but practical implementations lack sound open source solutions. The OneM2M-compliant platforms have proven capable of offering adequate and solid functionalities along with the desired openness and turned out to be successfully applicable to the control of optical networks.

In this work, we have investigated the performances of an OneM2M platform implemented through a Mobius server, and we have experimentally demonstrated that it can be a valid approach for the control and management of the considered optical networks without presenting significant shortcomings. The results obtained from our work show that the OneM2M platforms can scale well with an increasing workload and are flexible enough to build control applications operating in even complex networking scenarios. The OneM2M-based control architecture also showed reliability in the delivery of configuration messages, since all messages sent by the controllers successfully arrived at the ROADM agents within the prescribed timings in every considered set-up. However, the reliability of such platforms needs to be further investigated for larger networks or with more stringent control deadlines, along with the security aspects that have not yet been considered in this study.

As this work is among the first to our knowledge, it may pave the way for many other studies adopting the IoT approach as a paradigm for device-controller communication in the control plane of optical networks. Actually, the paper demonstrates that the optical network community could benefit from mature software platforms developed by an active open-source community operating in the IoT field.

## Figures and Tables

**Figure 1 sensors-22-00431-f001:**
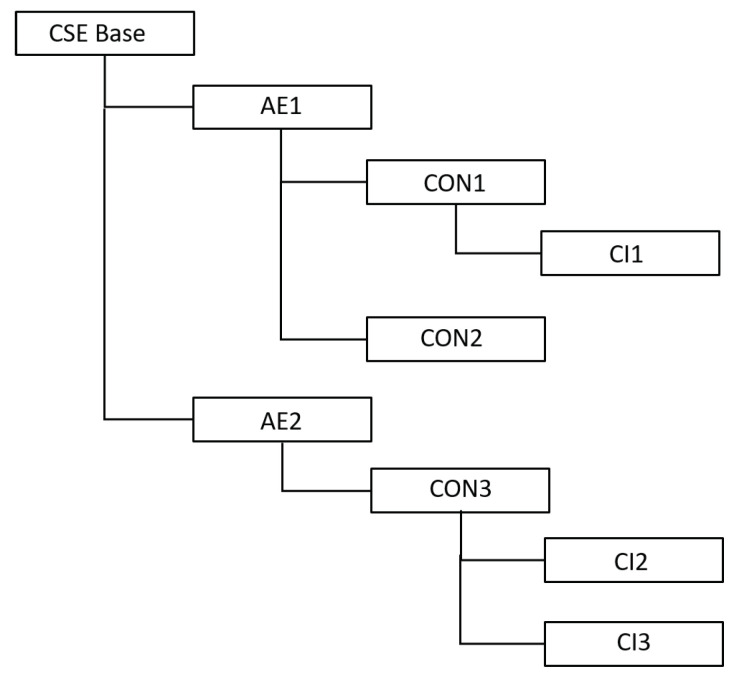
The OneM2M resource structure.

**Figure 2 sensors-22-00431-f002:**
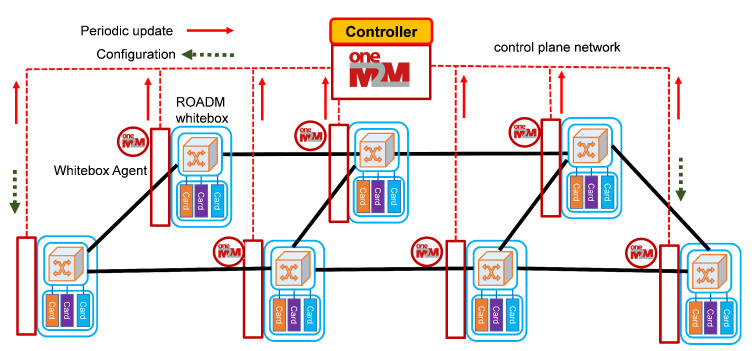
Reference scenario exploiting an OneM2M IoT platform for the SDN Control of optical networks.

**Figure 3 sensors-22-00431-f003:**
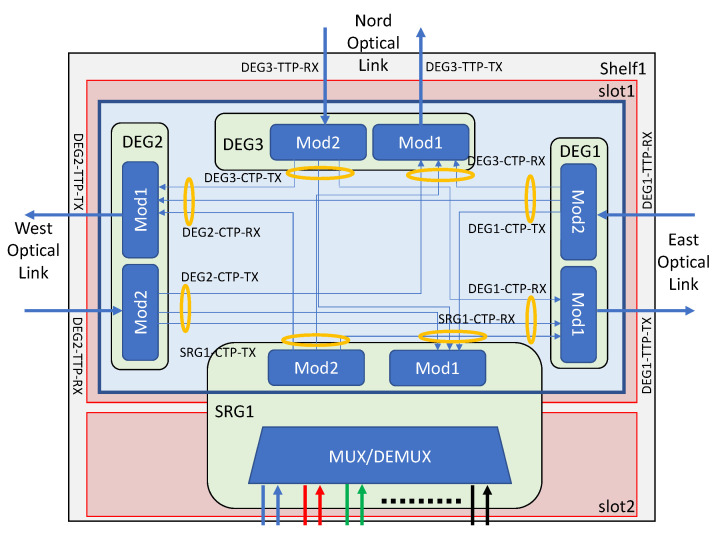
Example schema of an ROADM device.

**Figure 4 sensors-22-00431-f004:**
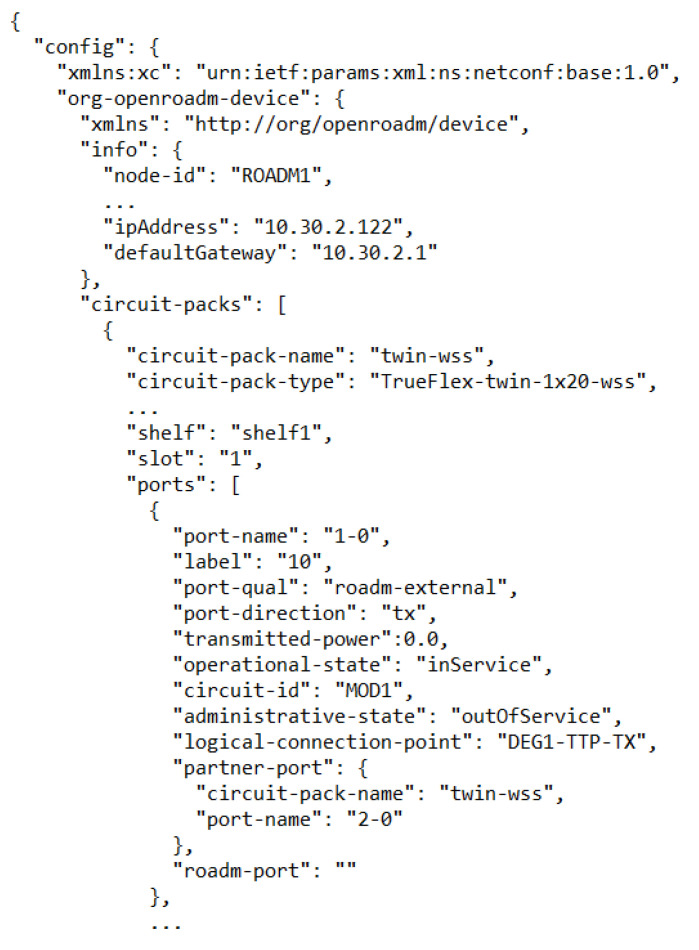
A piece of JSON message representing a model of an ROADM device.

**Figure 5 sensors-22-00431-f005:**
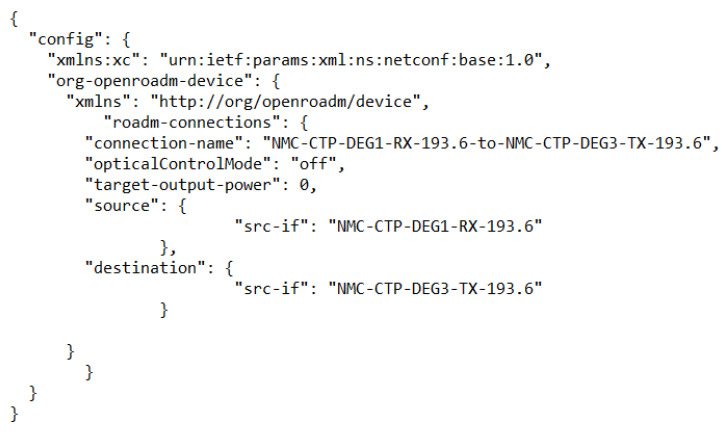
JSON model of a connection between ROADM devices.

**Figure 6 sensors-22-00431-f006:**
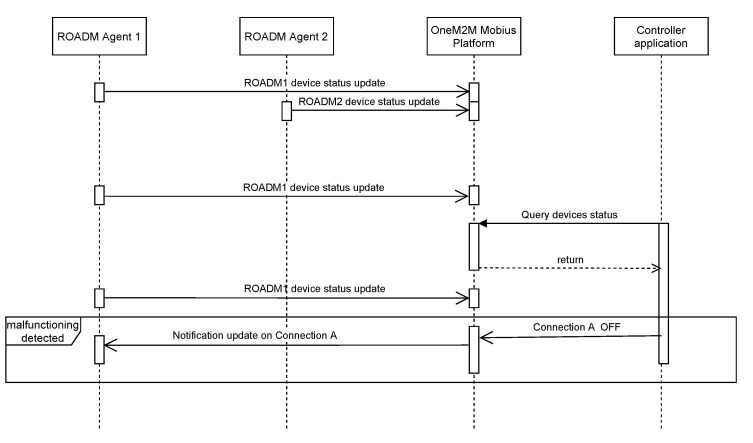
UML sequence schema of the interactions between the network components.

**Figure 7 sensors-22-00431-f007:**
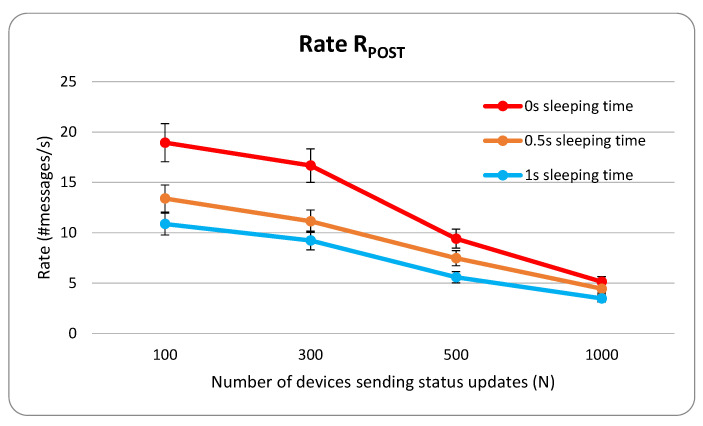
Representation of $R_{POST}$, varying the number of sending ROADM devices *N* and the sleeping time.

**Figure 8 sensors-22-00431-f008:**
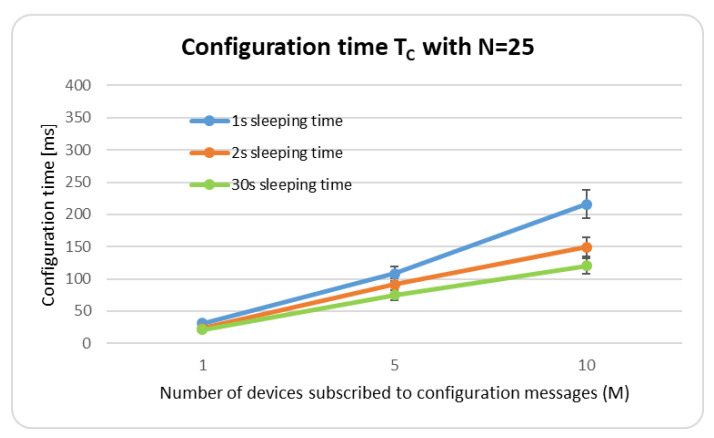
Representation of configuration time $T_{C}$, varying the number of agents subscribed to configuration messages *M* and the sleeping time between status updates of the *N* nodes.

**Figure 9 sensors-22-00431-f009:**
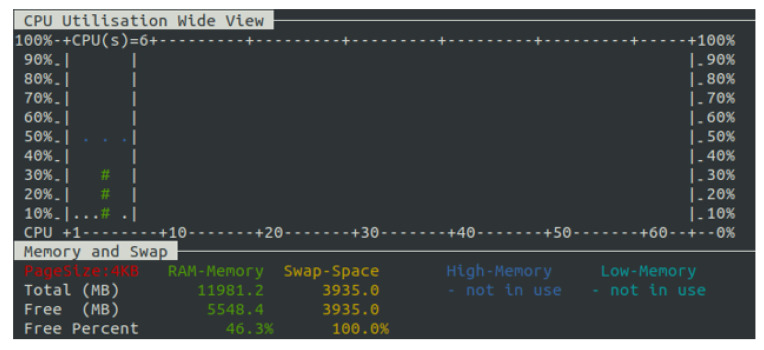
CPU and memory usage in the VM hosting the OneM2M platform.

**Figure 10 sensors-22-00431-f010:**
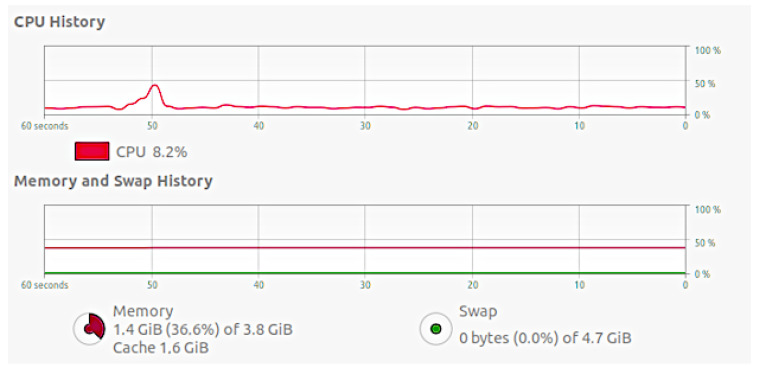
CPU and memory usage in one of the VMs hosting the ROADM agent.

**Table 1 sensors-22-00431-t001:** 25th percentile, median and 75th percentile of results for the first experiment.

Sleeping Time	25th Percentile	50th Percentile	75th Percentile
0 s	8.347222222	13.04166667	17.23611111
0.5 s	6.719097222	9.310833333	11.70861111
1 s	5.062430556	7.410694444	9.640625

**Table 2 sensors-22-00431-t002:** 25th percentile, median and 75th percentile of results for the second experiment.

Sleeping Time	25th Percentile	50th Percentile	75th Percentile
1 s	69.5	108	162
2 s	57	91	120
30 s	48	75	97.5

## Data Availability

The generated dataset is not publicly archived.

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
