# Peer review of "Scalable OneM2M IoT Open-Source Platform Evaluated in an SDN Optical Network Controller Scenario†"

_sensors, 2022, doi:10.3390/s22020431_

Round 1

Reviewer 1 Report

This paper experiments, assesses, and discusses the use of the OneM2M open-sourceplatform in the context of optical networks. Authors present valuable insight into the implementation of an IoT open-source platform applied to the management of optical networks. The experiment results show promising results and confirm that the OneM2M platform can be adopted in such a context.However, the following points should be addressed carefully before acceptance.

  1. The meaning of many English abbreviations in the paper is not clear enough.
  2. Please include relevant studies for 2021 in related work.
  3. In two experiments, the rate RPOSTand the time TCare considered separately. How are they determined? Any reference?
  4. The description of the results is relatively simple, and its internal reasons should be further analyzed and elaborated.For example, in Fig. 7, how are parameter status real-time updates to the controller represented? Why does the configuration time decrease as sleeping time increases in Fig. 8?
  5. English and writing style should be improved.

Reviewer 2 Report

In this article, authors try to experimentally evaluated the scalability of OneM2M platform with the connection to Software-Defined Optical Networks (SDON). I have following comments/suggestions/opinions with this article.

  • Authors are requested to highlight the contributions in points at the introduction section.
  • OneM2M platform’s connection to Optical Network particularly Software-Defined Optical Network (SDON) is still not clear, because OneM2M platform is for IoT, which is mostly to end-user/sensor network, since optical networks are backbone networks generally named with Transport-SDN. It is required to have clear explanation on Section 3.2 related to OneM2M platform connection with SDON.
  • Listing 1: the ROADM has only two modes: Tx or Rx, won’t be there an idle mode that it is neither sending nor receiving? Same applies to Listing 2.
  • L265: “….frequency 193.6THz and setting as launch power 0dBm..”. I wonder about the lunch power of 0dBm?? It means there is no lunch power! Please clarify the implication of having 0dBm or some power values greater than 0dBm.
  • Figure 6 looks like it is neither sequence nor interaction diagram, particularly the interaction with controller is not clear, please present it in standards UML diagram concerning communication with controller application. So that interactions among the network nodes shall be clearer.
  • What are the particular characteristics of Optical Network Controller (ONC)? Does this ONC has different characteristics than other normal controllers? Please elaborate on the features of ONC whether it is SDN based or not and its associations with OneM2M.
  • Since, scalability of OneM2M is evaluated from this article, but I wonder about the additional claim of high security and reliability of oneM2M platforms, indicated in L342 that a reference citation is required to state about OneM2M has high security and reliability.
  • Figure 3: is the ROADM device schema presented is your first work? Or if it is referred from other sources, please provide reference citation.
  • In Figure 4, for IoT or sensor network test/experimentation, it would be far better if authors consider IPv6 addressing/networking environment considering the latest generation networking instead of obsoleted IPv4 addressing environment.
  • L247: “Repeated experiment several times…” >> this means how many times? With varying sleep time. Is it possible to provide the dataset in annex about the test experiments? As you performed multiple stress test to observe behavior with different network load, which is good but let’s mention your test datasets generated if any.
  • Figure 8: for scalability evaluation, seems like 10 number of subscribed agents (M) and N=25 devices are not sufficient to evaluate the configuration time with respect to increasing number of devices. Maybe it needs more rounds of tests to justify your claim.
  • The scalability performance evaluation in terms of time and space complexity as a part of computation complexity still needs more experimentations over large network environment.
  • OneM2M has been used in several other scenarios, while authors n this article focused on its scalability evaluation with SDON. However the lack of theoretical/mathematical formulations in scalability evaluation and stick on to experimentation only constitute to weak presentation of this article. For the strength of this article to be strong enough for publication, I suggest authors to measure the computational complexity mathematically and experimentally/empirically both.

Reviewer 3 Report

  1. Three independent technologies OneM2M (IoT), SDN, and optical network are artifically integrated in this paper. The authors did not conduct indeepth analysis on what if it is not SDN, or what if it not optical network.
  2. The experiments are too simple. Major performance figures 8 and 9 are too simple to be even presented in a good conference. 
  3. The authors should make comparisons on performance for OneM2M with and without SDN, with and without optimal network. Why does narrowband IoT need SDN/optical that aim for broadband applications?
  4. Should consider advanced SDN platform such as Intel P4. We don't see how OneM2M affected the SDN code (such P4 math table and so on).
  5. The performance should be conducted in true measurements. The simulation approach simply cannot reflect the real situation in IoT/SDN.
  6. In conclusion, this paper is quite straightforward. Do not see much technical meats.

Round 2

Reviewer 2 Report

I did see the significant improvement in the manuscript.

May be authors have to change/rewrite the last paragraph of conclusion section where they surrender that it is just a preliminary study. Only simple preliminary study shall not be the strength for publication to Sensors. Hence, I request authors to rewrite last paragraph of conclusion and better include future works mentioning about what other researchers can follow with your preliminary study in the future. 
